

# Randomized clinical study of electrical impedance tomography-guided chest physiotherapy in difficult-to-wean patients: study protocol

Hao Wang[1], Jianing Xi[1] and Hongying Jiang[2]

[1] Beijing Rehabilitation Hospital, Capital Medical University, Beijing, China
[2] Department of Pulmonary and Critical Care Medicine, Beijing Rehabilitation Hospital, Capital Medical University, Beijing, China

## ABSTRACT

**Background**. Diaphragm dysfunction and inadequate chest wall and respiratory muscle function are common in critically ill patients who face difficulties in weaning from mechanical ventilation (MV). This can lead to secretion retention and impaired airway clearance. Chest physiotherapy (CPT) in these patients can help reduce secretion retention. This study protocol outlines an investigation into the feasibility and effectiveness of CPT guided by electrical impedance tomography (EIT) in difficult-to-wean patients on lung function, duration of mechanical ventilation, functional performance, and intensive care unit (ICU) hospitalization expenses.

**Methods**. This single-center, single-blind, randomized pilot study employed a parallel-group design. Participants will be randomly assigned to either an intervention group receiving EIT-guided CPT or a control group receiving only experiential CPT In-terventions, lasting 20 minutes each, were administered up to twice daily from ICU admission until discharge. The assessments will occur before interventions, after the 14 days, and after the 28 days. Outcome measures will be collected at before and after the intervention, peak expiratory flow, 30-day weaning success rate, maximum inspiratory pressure, diaphragm thickening rate, diaphragmatic excursion, ICU stay duration, and days from baseline to MV removal or Day 30. Adverse events will be documented.

**Conclusion**. This study will evaluate the feasibility and effectiveness of EIT-guided CPT versus conventional CPT in difficult-to-wean patients. If successful, this approach could enhance the efficiency of CPT.

**Trial registration**. This study was prospectively registered at the National Library of Medicine (https://clinicaltrials.gov/; reference: NCT06677099; Date of registration: 5th November 2024).

# INTRODUCTION

Mechanical ventilation (MV) is a critical life-support technique widely used in intensive care unit (ICU) settings, with 20–40% of patients requiring it daily. Despite its benefits, MV is associated with risks like ventilator-induced lung injury and pulmonary infection (*Wunsch,*

Corresponding authors
Jianing Xi, 790513822@qq.com
Hongying Jiang, bjkfdr@163.com

*Kramer & Gershengorn, 2017*). Patients who struggle to wean from MV often experience diaphragm dysfunction and weakened chest wall and respiratory muscles, resulting in small tidal volumes and ineffective coughing (*Slutsky & Ranieri, 2014*; *Timsit et al., 2017*). These issues increase the risk of retained airway secretions, which can lead to atelectasis and lower respiratory tract infections (LRTIs). Preventing or reversing these complications could reduce MV duration, mortality, and overall inpatient costs for difficult-to-wean patients (*Dasta et al., 2005*).

Previous research has demonstrated that chest physiotherapy (CPT) is essential for patients in long-term care, as it promotes mucociliary clearance, optimizes gas exchange, and reduces infections caused by mucostasis, ultimately enhancing quality of life (*Rosenthal et al., 2014*). Widespread CPT, including methods like postural drainage, percussion, vibration, and mechanical airway clearance, benefits critically ill patients by promoting lung recruitment, increasing lung volume and expiratory flow, and helping prevent and manage ventilator-associated pneumonia (*Longhini et al., 2020*).

However, traditional clinical observations and laboratory tests lack objective measures for monitoring changes in ventilation distribution and lung function during CPT. Consequently, the CPT plan is not fully personalized according to individual needs (*Spinou et al., 2024*).

Electrical impedance tomography (EIT) is an innovative, radiation-free imaging approach that can be employed at the bedside. It is based upon the ability to apply a weak current through a local electrode to characterize changes in thoracic bioelectrical impedance during breathing and utilize a corresponding imaging algorithm to monitor ventilation functionality across different lung regions, presented as real-time dynamic lung tomographic ventilation images (*Scaramuzzo et al., 2024*). In clinical practice, EIT is primarily used to assess patients' ventilation/perfusion status (*Frerichs et al., 2017*; *He et al., 2000*) and to evaluate treatment effectiveness (*Jiang et al., 2021*). Recent studies have explored EIT-guided ventilation strategies (*He et al., 2021a*; *He et al., 2021b*; *Hsu et al., 2021*) and used EIT to observe ventilation changes due to physiotherapy interventions (*Eimer et al., 2021*; *Yuan et al., 2021*). However, to date, no studies have investigated the individualized treatment strategies of using EIT to directly guide CPT.

This study aims to evaluate the feasibility of EIT-guided CPT and to analyze its effectiveness in difficult-to-wean patients with airway secretion retention and clearance impairment.

## METHODS

### Trial design and study setting

This single-center, single-blind, randomized pilot study employed a parallel-group design. This protocol adhered to the Standard Protocol Items: Recommendations for Interventional Trials (SPIRIT) guidelines (*Nydahl et al., 2017*) (see Table 1).

Inclusion criteria are mechanical ventilation patients who are ≥18 years old, received invasive mechanical ventilation more than 96 h before randomization, are satisfied the preconditions for the machine to be withdrawn, and at least have one failed attempt to
**Table 1 Recommendations for Standard Protocol Items: Recommendations for Interventional Trials (SPIRIT) guidelines.**

|  | Study period | | | | | |
| --- | --- | --- | --- | --- | --- | --- |
|  | Enrollment | Allocation | Post-allocation | | | Close-out |
| Timepoint** | $-t_1$ | $t_0$ | $t_1$ | $t_2$ | $t_3$ | $t_4$ |
| **ENROLLMENT:** |  |  |  |  |  |  |
| Eligibility screen | X |  |  |  |  |  |
| Informed consent | X |  |  |  |  |  |
| Allocation |  | X |  |  |  |  |
| **INTERVENTIONS:** |  |  |  |  |  |  |
| Control group |  |  |  |  |  |  |
| EIT-guided group |  |  |  |  |  |  |
| **ASSESSMENTS:** |  |  |  |  |  |  |
| **Baseline variables** |  |  |  |  |  |  |
| Age | X |  |  |  |  |  |
| Gender | X |  |  |  |  |  |
| BMI | X |  |  |  |  |  |
| APACHE-II | X |  |  |  |  |  |
| Mode of ventilation | X |  |  |  |  |  |
| PEEP | X |  |  |  |  |  |
| Duration of mechanical ventilation | X |  |  |  |  |  |
| Tracheostomy days | X |  |  |  |  |  |
| **Outcome variables** |  |  |  |  |  |  |
| Peak expiratory flow |  |  | X | X | X |  |
| The cumulativeincidence of successful weaning by Day 30 |  |  | X | X | X |  |
| Maximumin spiratory pressure |  |  | X | X | X |  |
| Diaphragm thickening rate |  |  | X | X | X |  |
| Breathing Frequency |  |  | X | X | X |  |
| Diaphragmatic excursion |  |  | X | X | X |  |
| Length of ICU stay |  |  | X | X | X |  |
| Cumulative incidence for death before successful weaning |  |  | X | X | X |  |
| The number of days from baseline to removal from MV as a result of successful weaning or Day 30 |  |  | X | X | X |  |
| Satisfaction with rehabilitation nursing |  |  | X | X | X |  |
| The number of adverse events during the entire period |  |  | X | X | X |  |

**Notes.**

SPIRIT schedule of enrolment, interventions, and assessments.

BMI, Body mass index; PEEP, positive end-expiratory pressure; APACHE-II, Acute Physiology and Chronic Health Evaluation.

withdraw the ventilator (re-need ventilator support within 48 h after extubation), and willing to participate in the study and sign the informed consent.

Exclusion criteria are as follows: malignant arrhythmia or acute myocardial ischemia; pneumothorax, pulmonary bulla and barotrauma and other lung diseases; hemorrhagic disease or abnormal coagulation mechanism with bleeding tendency; chest skin trauma; pulmonary hypertension and pulmonary embolism; with a permanent or temporary pacemaker; there is malignant tumor; present and previous history of neuromuscular diseases affecting respiratory muscle; participated in another clinical study related to

mechanical ventilation withdrawal; can not cooperate with the study for any reason or the researcher thinks that it is not suitable to be included in this experiment.

Only when we obtain the written consent before ICU admission can we verity the eligibility of the patients and start our trial process.

### Recruitment

The recruitment is mainly completed through the introduction of the ICU clinicians in our hospital and posters. For patients with severe or critical patients, we will work with their legal representatives to obtain permission.

### Ethics

Approval was given from our ethics committee (Ethics Committee of Beijing Rehabilitation Hospital of Capital Medical University, Agreement number 2022bkky-139). If adjustments or changes to the protocol are necessary, a project amendment will be requested from the committee. Before enrollment, patients are required to sign the informed consent form and agree to participate in the study.

### Interventions

Prior to the CPT plan being dispatched to centers, subjects in control group and EIT-guided group are given clinical treatment which is jointly formulated by the pulmonary rehabilitation team, and the course of treatment is determined according to the specific conditions of the patient. The medical and paramedical staff at each investigation center will be informed of the CPT plan and the use of EIT, in this way, make the teams can familiarize themselves with the technique. To limit operation bias, therapists should obtain training on standard operating procedures for EIT as well as a corresponding consistency assessment for EIT image evaluation results. Training on the standard operating procedure for EIT should include the correct use of EIT electrode strips, reduction of image artifacts during EIT, and correct understanding of the physiological significance of EIT ventilation area changes. The consistency assessment for EIT image evaluation should be conducted regularly (monthly), including experience sharing of clinical classic cases and detailed EIT use records of enrolled patients. All therapists participating in the study must have a minimum of three years of work experience.

Control group: Two sessions of CPT (morning and afternoon, 20 min each) are conducted. The CPT session consist of modified postural drainage (*Freitas et al., 2018*), assisted cough technique (*Spinou, 2020*), positive expiratory pressure (*McIlwaine, Button & Dwan, 2019*) and chest percussion, vibration (*Wilson, Morrison & Robinson, 2019*). The appointed therapist performed traditionally pulmonary auscultation and thoracic palpation to assess the status of pulmonary ventilation and secretion retention, and whether the patient's cough ability can complete effective airway clearance. Subsequently, CPT will be implemented according to the evaluation results and the experience of the therapist.

EIT-guided group: Similar to the patients in the control group, operation frequency and duration of CPT are predefined for the patients according to the assessment prior to the randomization. In addition to the traditional pulmonary auscultation and thoracic

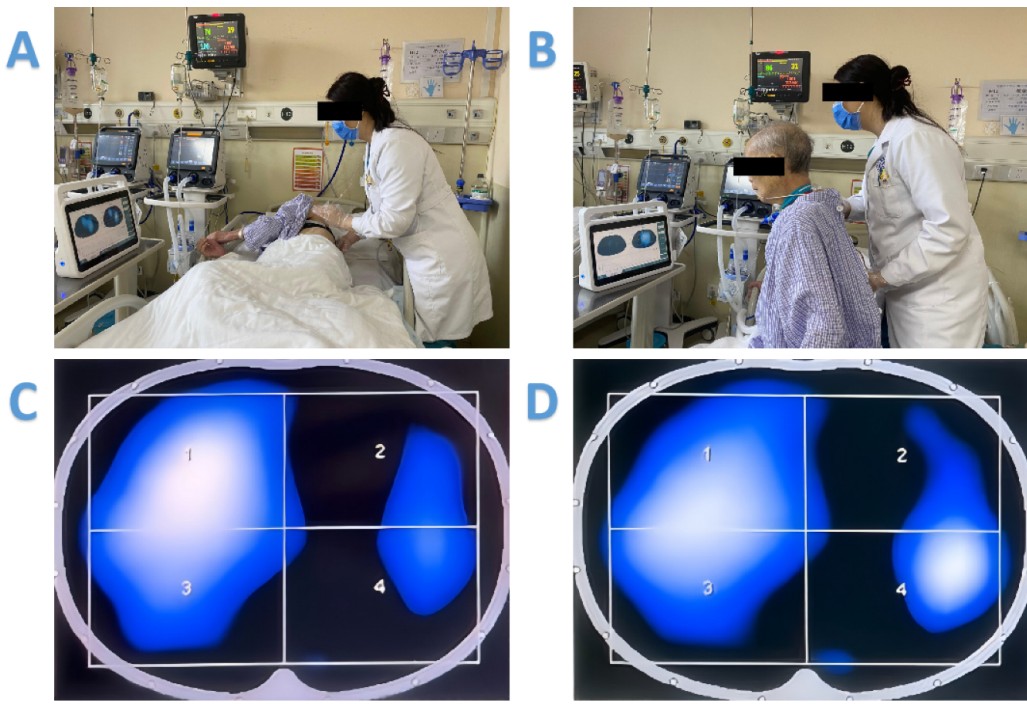

**Figure 1 The uses of EIT to guide individual treatments.** (A) EIT-guided modified postural drainage combined with vibrations and chest percussion. (B) EIT as a tool for instant feedback and motivation. (C) EIT images showed that ventilation on the left side is worse than that on the right side. (D) EIT images showed that after adjusting the patient's position to the right arm recumbent, and targeted vibration and percussion treatment in the left area of poor ventilation, symptoms improved significantly after this treatment, and left ventilation is better than before.

palpation, for each CPT session, EIT measurement is conducted and the images are used to guide the CPT treatments. The uses of EIT to guide individual treatments are briefly described as follows (Fig. 1).

EIT-guided modified postural drainage combined with vibrations and chest percussion: tidal variation images in EIT reveals heterogeneously ventilated regions. Physiotherapist identified such regions at the bedside and instructed the patient to take the appropriate drainage position, so that the poorly ventilated regions became gravity non-dependent regions. Subsequently, the physiotherapist put her hands on the poorly ventilated area with a vibratory force. A compressive pressure is produced by the therapist's arms, and the strength and therapeutic efficacy of CPT manipulation will be reported based on the EIT image results.

EIT as a tool for instant feedback and motivation: After each treatment session, the effect of the treatment is assessed immediately. The ventilation improvement is visible and explained using the EIT images. The patient visualized the improvement and is motivated for further CPT sessions.

## Outcomes (primary and secondary)

### Primary outcomes measure

The primary endpoint is assessing the peak expiratory flow through passive expiration, being considered the greatest value of the flow in the expiratory phase. All data will be recorded at the baseline (T1), 14 days (T2) and 28 days (T3).

### Secondary outcome measures. The secondary endpoints are:

- The accumulated incidence of successful weaning on the 30th day (note: successful weaning defined as "no need for reintubation within 48 hours" in patients without tracheostomy and "off the ventilator for 24 h and not reconnected to the ventilator within the next 48 hours" in patients with tracheostomy).
- Maximum inspiratory pressure (all data will be recorded at the baseline (T1), 14 days (T2) and 28 days (T3)).
- Diaphragm thickening rate (all data will be recorded at the baseline (T1), 14 days (T2) and 28 days (T3)).
- Diaphragmatic excursion (all data will be recorded at the baseline (T1), 14 days (T2) and 28 days (T3)).
- Length of ICU stay.
- ICU hospitalization expenses: including all medical expenses accumulated during the patient's ICU stay (the period spanning from study enrollment to ICU discharge).
- The number of days from baseline to removal from MV as a result of successful weaning or Day 30, whichever came first.
- Cumulative incidence for death before successful weaning.
- Patient satisfaction with rehabilitation nursing beginning at transfer from the ICU (the questionnaire related to patients' satisfaction with rehabilitation nursing is separated into five items, with five points for each item, for a total score of 25. The original questionnaire includes six items on which patients rated their satisfaction with respiratory rehabilitation on a scale of 1-strongly disagree to 5-strongly agree) (*Davies & Silvestre, 2020*).
- The number of adverse events during the entire period.

## Sample size

The sample size for this randomized clinical trial was determined utilizing G*Power software version 3.1, predicated on an effect size of $-0.62$ observed in change in right-side diaphragm thickening fraction, as detailed in previous research (*Dres et al., 2022*). To achieve a statistical power of 0.80 at an alpha level of 0.05, a total of 84 participants is requisite, allocating 42 individuals to each of the two comparative groups. The statistical analysis of the sample size calculation is conducted using the $t$-test for the difference between two independent means (two groups). Assuming a 20% drop-out rate, we ultimately decided to enroll a sample of 100 patients. However, after data is collected from the first 40 patients, an interim analysis will be performed to determine the size of the expected effect to be reasonable. Following that, a reassessment of the appropriate sample size will be performed based on this analysis.

## Randomization

Eligible patients will be randomly assigned (1:1) to either the EIT-guided or control group using a computer-generated block design managed by an independent operator. Treatment allocation will be concealed through sequentially numbered, opaque, sealed envelopes, ensuring that neither researchers nor participants are aware of group assignments. An impartial researcher, uninvolved in data collection, will oversee the randomization process.

## Blinding

An independent researcher, uninvolved in outcome evaluation, will assign group allocations. Blinding will extend to physiotherapists assessing outcomes. Participants and EIT-guided CPT providers will be instructed to avoid disclosing allocation to evaluators. The participants and researchers involved in the assessments will be blinded. Blinding will be maintained until data entry is completed, with any accidental unblinding will be documented. Outcome assessors will receive thorough training to ensure data quality.

## Data collection and management

To protect confidentiality, all real-time data will be coded, and random checks will be performed to ensure consistency across original records, forms, and database entries, with any discrepancies promptly corrected. Tomographic data will be securely stored alongside participant codes for future reference if needed. A comprehensive checklist will ensure strict adherence to research protocols, minimizing data omission or bias. In cases where missing data cannot be retrieved, affected participants will be excluded from the analysis. Data and analysis codes can be accessed by contacting the corresponding author. If any withdraw, we will perform an intention-to-treat analysis or imputation data.

## Statistical analysis

Normally distributed continuous data are described as means and standard deviation (SD) and non-normal distributed data are described as median (IQR). Normality of continuous data are assessed through the Shapiro–Wilk test. Comparisons between groups are performed by using the independent-samples t test or the Mann–Whitney U test, for the continuous variables normally or non-normally distributed, respectively. Categorical variables will be reported as numbers and percentages and compare with a Chi-square test or Fisher test. In addition, after the completion of the study, the sample can be divided into several subgroups for stratified analysis according to specific criteria (the type of primary disease, duration of mechanical ventilation, *etc.*). Primary and secondary endpoints will be evaluated using the modified intent-to-treat (mITT) population (the same as the ITT population for the control group but excluding patients for whom CPT plan is not achieved in the treatment group). We will use GraphPad Prism software (version 5.0) for the paper drawing, all statistical analyses is done with SPSS 22.0 (IBM, Armonk, NY, USA). Significant differences between groups or across time are reported at the alpha level of 0.05. All reported $p$ values are two-sided.

## Quality assurance

An external researcher uninvolved in measurements will perform random audits every 14 days to verify consistency between original equipment data, recorded formats, and the

database. Detected errors will be corrected promptly. The study's progress will be reviewed at monthly project meetings.

### Adverse events

In this trial, adverse events are defined as any unexpected medical occurrences presenting symptoms not observed before the study. Predicted adverse events include severe pain, removal of endotracheal, arterial, or central lines, arrhythmia, bradycardia, systolic blood pressure > 200 mmHg or < 90 mmHg due to hemodynamic instability during exercise, and desaturation ≤88% due to respiratory instability during exercise, along with minor reactions such as dyspnea, dizziness, tachypnea, and sinus tachycardia (*Schaller et al., 2016*; *Bailey et al., 2007*). Predicted side effects are adverse events, with severity categorized as mild, moderate, or severe. Both device-related and unrelated adverse events are recorded. In cases of unexpected serious adverse events or other unintended interventions, reports are submitted to the institutional review board. Data and safety monitoring will occur after every five participants are enrolled. In case of any harm related to the intervention, the researchers will provide all the necessary support.

## DISCUSSION

This study aims to assess the feasibility and effectiveness of EIT-guided CPT compared to conventional CPT in difficult-to-wean patients, who are at high risk for secretion clearance issues and have complex lung conditions that warrant precise and safe CPT techniques (*Martinez et al., 2022*).

The goals of CPT are to limit airway obstruction, improve ventilation and gas exchange, and improve respiratory muscle function. However, one primary challenge with current CPT practice is the absence of objective assessment of ventilation distribution (*He et al., 2021a*; *He et al., 2021b*). Our previous work has demonstrated that EIT can reliably assess ventilation improvement at the bedside in pneumonia patients (*Lim et al., 2009*). Inspired by these findings, we propose the use of EIT feedback to guide CPT, drawing on positive feedback from physiotherapists and patients. An individualized CPT program is formed according to the assessment, internal guidelines, and the patient's tolerance, education level, and patient preference. The most important step of EIT is to understand the cause of the current respiratory status. The spatial and temporal ventilation distribution of the patients often varies. After beginning the EIT measurement, poorly ventilated areas are characterized through ventilation distribution (*e.g.*, defect score (*Fratti et al., 2024*)). The potential causes of poorly ventilated are as are discussed with physicians, integrating other clinical evidence. Thereafter, the CPT program is designed or adjusted accordingly. Images are used to illustrate alterations in lung ventilation.

Notably, peak expiratory flow (PEF) was the primary outcome measure as some common indicators of clinical benefit, like weaning time or ICU stay length, are influenced by many factors. As an exploratory study, the primary goal was to investigate the effectiveness of EIT in informing CPT treatment. The major outcome measure was therefore selected surrounding a description of physiological benefit. Additionally, this study used 14 days and 28 days as critical timeframes for assessment. This is primarily due to the longer

duration of mechanical ventilation in the weaning population in this study, possible delays in decannulation after the first SBT, and influences of reintubation after decannulation on outcomes (*Qian et al., 2022*). An additional extended observation period of 30 days made the findings more robust.

This pilot randomized controlled trial is the first, to our knowledge, to systematically evaluate a bedside feedback approach for guiding CPT in difficult-to-wean patients. EIT's low cost and high sensitivity to thoracic air content changes make it a promising tool for individualized CPT planning, allowing physiotherapists to tailor treatments to each patient's ventilation profile and adjust as needed. If the study hypothesis is confirmed, these findings will change and standardize rehabilitation nursing practices and make the management of difficult-to-wean patients in intensive care more rational.

However, this protocol has some limitations that must be acknowledged. This procedure will be performed initially in only one center. In the future, a multi-center trial will strengthen the external adaptability of the findings. Furthermore, the ICU presents a mixed profile of patients, including differences in clinical treatment and mechanical ventilation duration prior to enrollment.

### Funding
The authors received no funding for this work.

### Competing Interests
The authors declare there are no competing interests.

### Author Contributions
- Hao Wang conceived and designed the experiments, performed the experiments, analyzed the data, prepared figures and/or tables, authored or reviewed drafts of the article, and approved the final draft.
- Jianing Xi analyzed the data, prepared figures and/or tables, and approved the final draft.
- Hongying Jiang conceived and designed the experiments, prepared figures and/or tables, authored or reviewed drafts of the article, and approved the final draft.

### Human Ethics
The following information was supplied relating to ethical approvals (*i.e.*, approving body and any reference numbers):

Ethics Committee of Beijing Rehabilitation Hospital of Capital Medical University (Agreement number 2022bkky-139).

### Clinical Trial Ethics
The following information was supplied relating to ethical approvals (*i.e.*, approving body and any reference numbers):

Chinese Clinical Trial Registry (https://www.chictr.org.cn/); National Library of Medicine (https://clinicaltrials.gov/).

## Data Availability

This is a registered report.

## Clinical Trial Registration

The following information was supplied regarding Clinical Trial registration:

ChiCTR2400087759; NCT06677099

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
