# Peer review of "Randomized clinical study of electrical impedance tomography-guided chest physiotherapy in difficult-to-wean patients: study protocol"

_PeerJ, doi:10.7717/peerj.19727_

## Round 0.1 · original submission · Major Revisions

Dear Authors
Although this study protocol projects a great idea there text lack detailed explanation on how Electrical Impedance Tomography Guided Chest Physiotherapy is delivered. In addition this is a study protocol but in multiple places the article is written in past tense which makes me concerned if you have already if this is a protocol of a study that is already completed. One of the reviewers has also raised a concern about the date of registration of the trial. Kindly address all the concerns raised by the reviewers and submitted an updated manuscript.

·

Basic reporting

Manuscript was written well.
More and updated references are needed.
Background/rationale/gap/need/significance of study are insufficient...identify the problem/gap (comment on other available methods to deal with the problem, what you will solve/add...etc), show mechanism of EIT, how it will solve the problem, how and whom this will benefit.

Experimental design

Article follows the scope of the journal.
The research question was defined well.
The gap is lacking.
Methods need more details.. e.g sample size (effect sized?? based on what??)...etc. (more details were commented on the attached file).

Validity of the findings

Not applicable.

Additional comments

More comments were present in the attached file.

Reviewer 2 ·

Basic reporting

Strengths of the protocol:

1. Use of Innovative Technology (EIT):
- The use of Electrical Impedance Tomography (EIT) as a tool to guide chest physiotherapy is a highlight, allowing real-time assessment of ventilation and adaptation of treatment to the specific needs of each patient. This improves the accuracy and efficacy of treatment.

2. Study Design:
- The study presents a randomized design, with parallel groups and blinding of evaluators, which increases the validity of the results and reduces biases.

3. Well-defined inclusion and exclusion criteria:
- Clearly defined criteria ensure that the sample is representative of the target group, increasing the clinical relevance of the results.

4. Comprehensive primary and secondary outcomes:
- The selected outcomes encompass a robust assessment of efficacy, including physiological measures, weaning success rate, and length of ICU stay.

5. Strict Quality and Safety Control:
- Implementation of regular audits and training to prevent protocol deviations and ensure data quality.

Weaknesses of the protocol:

1. Sample Size:
- Although it justifies the calculation, the sample of 80 participants may be insufficient to detect statistically significant differences in secondary or rare outcomes.

2. Partial Blinding:
- Although the evaluators are blind, the participants and professionals conducting the physical therapy are not. This may introduce behavioral and performance biases.

3. Generalizability of Results:
- Because the study was conducted at a single center, the applicability of the results to other clinical settings may be limited.

4. Dependency on Specific Equipment:
- The use of EIT may limit replication in centers that do not have access to this technology.

Possible solutions to the weaknesses:

1. Sample Expansion:
- Increase the sample size to include multiple centers, ensuring greater statistical power and generalizability.

2. Extensive Training and Additional Audits:
- Improve training to minimize the impact of partial blinding, ensuring uniformity in treatment between groups.

3. Multicenter Trials:
- Expand the study to multiple centers, which will increase the representativeness of the sample and the applicability of the results.

4. Alternatives to EIT:
- Consider complementary assessment methods that can be used in settings where EIT is not available, to expand the clinical impact.

These suggestions aim to optimize the scientific robustness of the protocol and increase the clinical impact of the findings.

Experimental design

no comment

Validity of the findings

no coment

·

Basic reporting

• Language and Presentation: The manuscript is well-written, but there are minor grammatical errors (e.g., “consentsent” in inclusion criteria) and occasional awkward phrasing that should be revised for clarity and readability.
• Structure and Clarity: The manuscript adheres to standard formatting for protocol studies, with clearly labeled sections and a logical flow. Figures or diagrams illustrating the EIT-guided intervention could enhance clarity.
• References: The references are relevant and up-to-date; however, some inconsistencies in formatting should be standardized according to journal guidelines.
• Context and Background: The introduction provides a thorough overview of the clinical problem, the rationale for the study, and the novelty of using EIT to guide chest physiotherapy.

Experimental design

• Study Design: The study employs a robust single-blind, randomized, parallel-group design appropriate for addressing the research question. The inclusion and exclusion criteria are well-defined, ensuring the selection of a relevant patient population.
• Intervention Description: The EIT-guided intervention is innovative, but the protocol would benefit from additional clarity. For instance, more detail on how EIT results influence the physiotherapy approach or the inclusion of a flowchart or diagram would make the methodology more accessible.
• Control Group: While the control group undergoes conventional CPT, the manuscript should explain how treatment variability among control patients will be minimized or standardized.
• Outcome Measures: The primary and secondary outcomes are clearly defined and clinically relevant. However, the manuscript could elaborate on the choice of time points for measurements and their significance in evaluating patient recovery.
• Sample Size: The sample size calculation is briefly described, but more detail on the assumptions (e.g., expected effect size, standard deviation) would enhance the rigor of this section.

Validity of the findings

• Feasibility: The protocol is feasible within the described single-center ICU setting, and the study has been designed to minimize bias through randomization, blinding, and independent outcome assessments.
• Statistical Analysis: The statistical methods are appropriate for the study’s objectives. However, handling potential confounders, missing data, and sensitivity analyses for adverse events should be discussed in more detail.
• Ethical Considerations: The manuscript adheres to ethical guidelines, including informed consent, data confidentiality, and adverse event management. The protocol for handling severe or unexpected adverse events could be expanded for greater transparency.
• Trial Registration: The trial is registered, but the listed registration date (November 5, 2024) appears to be in the future and requires clarification.

Additional comments

• This study addresses an important issue in critical care and proposes an innovative solution using EIT to individualize CPT for patients who are difficult to wean from mechanical ventilation.
• While the manuscript is comprehensive, some sections could benefit from greater clarity and depth:
- Include a rationale for time points used in outcome measurement.
- Provide details on managing potential protocol deviations or missing data.
- Consider the addition of visual aids to illustrate the EIT-guided approach and patient pathway.
• Minor revisions to language and formatting are needed for consistency and readability.

---

## Round 0.2 · Minor Revisions

Dear Authors

One of the reviewers has recommended accepting the paper while the 2nd has recommended minor corrections. Kindly make the changes suggested by the reviewer and resubmit.

·

Basic reporting

Only some typographical errors e.g. Mechanical in abstract (small letter instead of capital M). As well, some undue spaces before and between words are present.

Experimental design

No comment

Validity of the findings

No comment

Reviewer 2 ·

Basic reporting

.

Experimental design

.

Validity of the findings

.

Additional comments

Reviewer Comments – PeerJ
Manuscript Title: Randomized Clinical Study of Electrical Impedance Tomography-guided Chest Physiotherapy in Difficult-to-Wean Patients: Study Protocol

General Assessment
This study protocol presents a well-conceived and innovative randomized controlled pilot trial that evaluates the use of Electrical Impedance Tomography (EIT) to guide chest physiotherapy (CPT) in difficult-to-wean patients in the ICU. The topic is highly relevant and addresses a clear clinical gap, the lack of objective, individualized monitoring tools for optimizing CPT in mechanically ventilated patients.
The rationale is sound, the intervention is clearly described, and the outcomes are relevant and well-aligned with the study objectives. The manuscript is generally well-written and structured according to SPIRIT guidelines. If successfully completed, this study has the potential to influence and advance rehabilitation practices in intensive care.
However, several methodological and conceptual aspects would benefit from clarification or improvement prior to publication of this protocol. Below, I outline the strengths, limitations, and specific recommendations.

Strengths
1. Originality and Innovation
o This is the first RCT, to the best of my knowledge, evaluating real-time, EIT-guided CPT in this patient population. The individualized approach and visual feedback for both clinicians and patients are novel and potentially impactful.
2. Well-grounded rationale
o The background section is comprehensive and references high-quality, recent evidence supporting the proposed intervention.
3. Robust trial design
o Randomized, single-blind design with clearly defined inclusion/exclusion criteria and use of allocation concealment.
4. Appropriate and diverse outcome measures
o The combination of physiological, clinical, and patient-centered endpoints (e.g., peak flow, inspiratory pressure, diaphragm function, satisfaction) offers a well-rounded evaluation of the intervention's effectiveness.
5. Statistical planning
o Adequate sample size estimation based on prior data, with provision for interim analysis and dropout.

Major Concerns and Recommendations
1. Justification of Primary Outcome
o While peak expiratory flow is physiologically relevant, it may not fully capture clinical impact. The authors are encouraged to justify its selection as the primary endpoint over more directly clinical measures such as time to successful weaning or ICU length of stay.
2. Blinding and Performance Bias
o Although outcome assessors are blinded, physiotherapists administering EIT-guided CPT are necessarily unblinded. The authors should discuss how they plan to mitigate potential performance bias arising from this unblinding.
3. Operator Experience and Standardization of EIT Use
o Given that EIT interpretation is partly subjective, the authors should describe any training or competency assessment provided to the physiotherapists using EIT. This would improve confidence in intervention fidelity.
4. Patient Engagement and Motivation
o The use of EIT images as real-time feedback for patients is intriguing. Consider incorporating a validated tool to assess patient experience or motivation, or at minimum, provide more detail on how this aspect will be measured or qualitatively assessed.
5. Health Economic Evaluation
o The study mentions hospitalization expenses as a secondary outcome, but no methodology for cost analysis is provided. The authors should specify which costs will be tracked (e.g., ICU stay, interventions, medications), and how they will be analyzed.
6. Generalizability
o The single-center design and heterogeneity of ICU patients are acknowledged limitations. It would be helpful to briefly discuss plans (if any) for multicenter validation in a future phase.
7. Inter-rater Reliability for EIT Interpretation
o If possible, include a plan to assess inter-observer agreement in EIT image interpretation to ensure standardization of CPT guidance.

Minor Suggestions
• Improve clarity in some parts of the text (e.g., definition of “experiential CPT” in the control group).
• Consider rewording some grammatical issues, particularly in the abstract (e.g., "experiential CPT interventions" vs. "standard CPT interventions").

Final Recommendation
Accept with minor revisions.
This is a highly promising and well-structured protocol that addresses a novel and clinically important question. With minor clarifications and expansions on the points noted above, the protocol would be suitable for publication in PeerJ and would provide a valuable contribution to the field of critical care rehabilitation.

---

## Round 0.3 · accepted · Accept

Dear Authors
Thanks for making all the recommended changes.
Congratulations !

Reviewer 2 ·

Basic reporting

.

Experimental design

.

Validity of the findings

.

Additional comments

See attachment.
The authors have adequately addressed all concerns raised by Reviewer #2. All methodological, statistical, and procedural clarifications were incorporated into the revised manuscript. I recommend acceptance after minor English language polishing.

Annotated reviews are not available for download in order to protect the identity of reviewers who chose to remain anonymous.